# Novel Mutation in the Feline *GAA* Gene in a Cat with Glycogen Storage Disease Type II (Pompe Disease)

**DOI:** 10.3390/ani13081336

**Published:** 2023-04-13

**Authors:** Tofazzal Md Rakib, Md Shafiqul Islam, Shigeki Tanaka, Akira Yabuki, Shahnaj Pervin, Shinichiro Maki, Abdullah Al Faruq, Martia Rani Tacharina, Osamu Yamato

**Affiliations:** 1Laboratory of Clinical Pathology, Joint Faculty of Veterinary Medicine, Kagoshima University, Kagoshima 890-0065, Japan; rakibtofazzal367@gmail.com (T.M.R.); si.mamun@ymail.com (M.S.I.); yabu@vet.kagoshima-u.ac.jp (A.Y.); s.pervin30@yahoo.com (S.P.); k6993382@kadai.jp (S.M.); faruqabdullahal103@gmail.com (A.A.F.); martia.rt@fkh.unair.ac.id (M.R.T.); 2Faculty of Veterinary Medicine, Chattogram Veterinary and Animal Sciences University, Khulshi, Chattogram 4225, Bangladesh; 3Alpha Animal Hospital, Kawanakajima, Nagano 381-2226, Japan; alphaah@sea.plala.or.jp; 4Faculty of Veterinary Medicine, Airlangga University, Mulyorejo, Surabaya 60115, Indonesia

**Keywords:** Pompe disease, *GAA* gene, lysosomal disease, glycogen storage disease type II, cat

## Abstract

**Simple Summary:**

Glycogen storage disease type II (Pompe disease: PD) is an autosomal recessive metabolic disorder caused by mutations of the *GAA* gene encoding lysosomal acid α-glucosidase. Here, we studied the molecular basis of an eight-month-old domestic short-haired cat previously diagnosed with PD. Sanger sequencing was performed on 20 exons of the feline *GAA* gene using genomic DNA extracted from the paraffin-embedded tissues of this cat. A homozygous missense mutation (*GAA*:c.1799G>A, p.R600H) was identified as a candidate pathogenic mutation. Several stability and pathogenicity predictors showed that this mutation is deleterious and severely decreases the stability of acid α-glucosidase. The clinical outcomes and identified mutation were identical to those observed in human infantile-onset PD. This is the first report of a pathogenic mutation in the feline *GAA* gene.

**Abstract:**

Glycogen storage disease type II (Pompe disease: PD) is an autosomal recessively inherited fatal genetic disorder that results from the deficiency of a glycogen hydrolyzing enzyme, acid α-glucosidase encoded by the *GAA* gene. Here, we describe the molecular basis of genetic defects in an 8-month-old domestic short-haired cat with PD. The cat was previously diagnosed with PD based on the clinical and pathological findings of hypertrophic cardiomyopathy and excessive accumulation of glycogen in the cardiac muscles. Sanger sequencing was performed on 20 exons of the feline *GAA* gene using genomic DNA extracted from paraffin-embedded liver tissues. The affected cat was found to be homozygous for the *GAA*:c.1799G>A mutation resulting in an amino acid substitution (p.R600H) of acid α-glucosidase, a codon position of which is identical with three missense mutations (p.R600C, p.R600L, and p.R600H) causing human infantile-onset PD (IOPD). Several stability and pathogenicity predictors have also shown that the feline mutation is deleterious and severely decreases the stability of the GAA protein. The clinical, pathological, and molecular findings in the cat were similar to those of IOPD in humans. To our knowledge, this is the first report of a pathogenic mutation in a cat. Feline PD is an excellent model for human PD, especially IOPD.

## 1. Introduction

Glycogen storage disease type II (MIM # 232300), also known as Pompe disease (PD), is a rare autosomal recessive metabolic disorder caused by the deficiency of the lysosomal acid α-1,4 glucosidase (GAA, EC 3.2.1.20) [1,2]. Because of GAA deficiency, glycogen accumulates in the lysosomes of many tissues, particularly cardiac and skeletal muscles [3,4,5]. The human *GAA* gene is located on chromosome 17q25.3 and is approximately 20 kb in size. The feline *GAA* gene is located on chromosome E1. It contains 20 exons, of which exon 1 is non-coding and codes for 952 amino acids in both humans and cats [6,7].

In humans, PD can be classified as infantile-onset PD (IOPD) and late-onset PD (LOPD) [6]. The clinical symptoms of IOPD include generalized muscle weakness, respiratory distress, hypertrophic cardiomyopathy, hypotonia, and macroglossia [6,7]. Widespread muscular glycogenosis, cardiac hypertrophy, and hepatomegaly have been reported as pathological findings of IOPD [8,9,10,11]. The more common form, LOPD, has a heterogeneous presentation that occurs from childhood to late adulthood and is characterized by slowly progressive axial and/or limb-girdle muscle weakness, with or without respiratory symptoms. Sequestered respiratory failure, increased serum creatine kinase activity, atrial aneurysm, oropharyngeal dysphagia, ptosis, and scoliosis have also been reported in humans with LOPD [3,10].

To date, more than 900 mutations in the human *GAA* gene have been reported in the PD *GAA* variant database [12,13]. PD has also been described in other species, including dogs (OMIA 000419-9615) [1,14], cats (OMIA 000419-9685) [15], cattle (OMIA 000419-9913) [16,17], sheep (OMIA 000419-9940) [18], and the Japanese quail (OMIA 000419-93934) [19]. In dogs, a nonsense mutation, c.2237G>A (p.W746*), has been reported in Finnish and Swedish Lapphunds and was also reported at the same amino acid position in human IOPD [1]. Three pathogenic mutations: c.1057_1058del (p.Y353L), c.1783C>T (p.R595*), and c.2454_2455del (p.T819R), have been reported in Brahman and Droughtmaster, Brahman, and Shorthorn cattle, respectively [20]. To date, no mutations have been identified in cats, quail, or sheep with PD. Feline PD was first reported in a cat in 1969; however, the described pathological changes were limited to the brain because of the unavailability of liver and cardiac muscle specimens [1,15]. Molecular characterization was not performed at that time. The genetic defects underlying feline PD remain unknown.

Recently, our research group reported an eight-month-old domestic short-haired (DSH) cat with heart failure as another case of feline PD [11]. This study aimed to identify the feline pathogenic *GAA* mutation in this cat with PD.

## 2. Materials and Methods

This study was performed in accordance with the Guidelines Regulating Animal Use and Ethics of Kagoshima University (no. VM15041; approval date: 29 September 2015). Informed oral consent was previously obtained from the owner of the cat with PD [11].

### 2.1. Specimen

For DNA sequencing analysis, specimens were obtained from the paraffin-embedded liver samples from a DSH cat diagnosed with PD based on clinical, pathological, and ultrastructural findings [11]. Genomic DNA was extracted from the specimens using automated extraction equipment (magLEAD 6gC, Precision System Science, Co. Ltd., Matsudo, Japan).

### 2.2. Mutation Analysis

The coding exons and splice junctions of the feline *GAA* gene were amplified and assessed with polymerase chain reaction (PCR) and Sanger sequencing with specific primer pairs. The primer pairs were designed based on the reference sequence (XM_006940652.3) as the need arose because the DNA from the paraffin-embedded samples was modestly fragmented (sequences in Appendix A). PCR was performed in a 20 µL reaction mixture containing 10 µL 2X PCR master mix (GoTaq Hot Start Green Master Mix, Promega Corp., Madison, WI, USA). The PCR products were purified using a QIAquick Gel Extraction Kit (Qiagen, Hilden, Germany) according to the manufacturer’s instructions before sequencing. Sanger sequencing was performed by Kazusa Genome Technologies Ltd. (Kisarazu, Japan). The obtained sequencing data were compared with a reference sequence (XM_006940652.3) to identify candidate pathogenic mutations.

### 2.3. Pathogenicity and Stability Prediction

Pathogenicity and mutational changes in the stability of the predicted protein were analyzed using the PredictSNP, iStable, and FoldX servers, respectively. The PredictSNP server has an embedded algorithm for PredictSNP, MAPP, PhD-SNP, Polyphen-1, Polyphen-2, SIFT, SNAP, and Panther to classify the variants as ‘deleterious’ or ‘neutral’ [21]. Similarly, iStable has an embedded algorithm of iStable, i-Mutant, and MUpro to classify the variants as ‘increasing’ the stability and ‘decreasing’ the stability of the protein [22]. Additionally, the empirical protein design forcefield FoldX was used to calculate the difference in free energy of the mutation: delta delta G (ddG). Stability was predicted based on the ddG value. If a mutation destabilizes the structure, ddG increases, whereas stabilizing mutations decrease ddG. Since the FoldX error margin is around 0.5 kcal/mol, changes in this range are considered insignificant [23]. The amino acid sequences in the FASTA format and mutations were used as inputs for prediction.

### 2.4. PCR-Restriction Fragment Length Polymorphism (RFLP)

The PCR-RFLP was carried out with forward (5′-AGGGCCCTGGTCAAGGC-3′) and reverse (5′-TGACAGGCGCTCTCACCT-3′) primers in a 20 µL reaction mixture containing 10 µL 2X PCR master mix (GoTaq Hot Start Green Master Mix, Promega Corp), 12.5 pmol of each primer and extracted DNA of the affected cat and clinically healthy control cats as templates. After the first denaturation at 95 °C for 2 min, 40 cycles of amplification were carried out at an annealing temperature of 58 °C for 30 s and a final extension at 72 °C for 2 min. The PCR products were digested with the restriction enzyme *Aci*I (New England Biolabs, Ipswich, MA, USA) at 37 ℃ for 1 h in a 10 µL reaction mixture containing 8 µL of PCR product, 10 U of *Aci*I, and 2 µL of 10X restriction enzyme buffer. The digested PCR products were then visualized with agarose gel electrophoresis using 3% agarose gel using a molecular size marker (Marker 9, Nippon Gene Co. Ltd., Tokyo, Japan). One hundred DNA samples from mixed-breed cats, which were previously stored in the laboratory [24], were genotyped using PCR-RFLP.

## 3. Results

### 3.1. Mutation Detection

Sanger sequencing was performed on 20 exons and splice junctions of the *GAA* gene in the cat with PD. The sequenced exon and splice junctions were compared to the reference sequence (XM_006940652.3). As a result, the sequencing and comparison revealed 21 homozygous coding, two heterozygous coding, one splice junction, two intronic, and two untranscribed region (UTR) variants (Figure 1). The coding variants included 10 homozygous missense mutations (c.55G>A, c.181G>A, c.1298G>A, c.1480G>A, c.1779G>A, c.1889A>G, c.2152G>T, c.2207G>A, c.2423C>G, and c.2629G>A), one heterozygous missense (c.2728G>A), and 12 silent mutations (c.72T>C, c.840C>T, c909C>T, c.1200G>T, c.1350C>T, c.1560A>G, c.1710T>C, c.1779G>T, c.2178T>C, c.2304C>T, c.2373C>T, and c.2688G>A).

### 3.2. Pathogenicity and Stability Prediction

The 11 missense mutations were further analyzed using the SIFT and PredictSNP servers. One of the eleven identified missense mutations (c.1799G>A, p.R600H, Figure 2) showed a deleterious effect on the catalytic site of the GAA protein in the analysis of all eight pathogenicity predictors (Table 1). The SIFT prediction server also supported the same amino acid mutation (p.R600H) in exon 13 as deleterious, with a SIFT score (0.00). Another candidate mutation (c.55G>A, p.V19M) in exon 2 was predicted to be deleterious by five predictors with a moderately low SIFT score (0.03). These two predicted deleterious mutations were subjected to the iStable server to measure the change in the stability of the GAA protein upon mutation using the I-mutant2.0, MUpro, and iStable algorithms (Table 2). Only one mutation (c.1799G>A, p.R600H) in exon 13 was predicted to decrease the stability of the GAA protein by all three stability predictors.

To date, no model for the feline GAA protein crystal structure is available in the Protein Data Bank (PDB) database. Therefore, we performed a protein–protein BLAST analysis between the reference feline GAA protein sequence and the PDB database at the National Center for Biotechnology Information server to predict identical proteins. An identical human GAA protein (PDB ID: 5KZW, query coverage: 91%) was predicted and used to predict the stability in the FoldX predictor because of the unavailability of a feline GAA protein model in the PDB database. After FoldX prediction, the mutation from arginine to histidine at amino acid position 600 resulted in a ddG of 12.62 kcal/mol. This implied that this mutation (p.R600H) severely reduced GAA protein stability (Figure 3). However, the stability effect of the p.V19M mutation could not be predicted because of the absence of the first 79 amino acids in the reference protein model. Overall, the mutation c.1799G>A (p.R600H) in exon 13 was predicted to be deleterious and to decrease the stability of the GAA protein by all pathogenicity and stability predictors used in this study.

### 3.3. PCR-RFLP and Genotyping

Using the PCR-RFLP, stored control DNA samples from 100 clinically healthy cats were genotyped. All control samples were homozygous for the wild-type genotype (c.1799G/G), whereas only the cat with PD was homozygous for the mutant genotype (c.1799A/A) (Figure 4). We performed Sanger sequencing on five control DNA samples randomly selected from among 100 clinically healthy cats and found that three homozygous wild-type (c.55G/G), two heterozygous (c.55G/A), and one homozygous mutant (c.55A/A) genotypes.

## 4. Discussion

We reported an eight-month-old DSH cat with heart failure as the second case of feline PD in 2022 [11], after the first case was reported in 1969 [15]. The diagnosis in our cat was established based on clinical and pathological characteristics similar to those of human PD [11]. Our cat first presented at eight months of age with acute tachypnea, poor growth, hypothermia, and lethargy at a veterinary hospital after a three-month history of unexplained fever. Radiography revealed that the cat had cardiomegaly. Echocardiography revealed dilatation of both atria and left ventricular systolic dysfunction. The cat died of pulmonary edema caused by chronic heart failure. The severe clinical manifestations in the cat resembled those of human IOPD. Postmortem histological examination revealed severe vacuolation of the cardiac muscle cells stained with hematoxylin and eosin. Periodic acid-Schiff staining positively stained coarse granules within the vacuoles that disappeared upon pre-digestion with diastase, indicating that glycogen accumulated in the cardiac muscle cells. To identify the first pathogenic mutation in the feline *GAA* gene, we molecularly analyzed the genomic DNA from this cat with PD because the molecular basis of the genetic defect in feline PD remains unknown.

Sanger sequencing of the feline *GAA* gene of the affected cat and comparison with the reference sequence was performed to identify candidate pathogenic mutations in feline PD. As a result, 10 homozygous and 1 heterozygous missense mutations were found in the coding region of the feline *GAA* gene (Figure 1). After the evaluation of pathogenicity and stability prediction (Table 1 and Table 2 and Figure 1 and Figure 3), two missense mutations, c.1799G>A (p.R600H) and c.55G>A (p.V19M), were selected as the most likely pathogenic mutations in the cat with PD. Based on the data obtained from several stability and pathogenicity predictors, the c.1799G>A mutation was more likely to be a pathogenic mutation. Furthermore, a preliminary genotyping survey strongly suggested that c.1799G>A is a pathogenic mutation in feline PD. That is because there were no cats with the c.1799G>A mutation in the 100 clinically healthy cats, whereas 2 heterozygous and mutant homozygous cats for the c.55G>A mutation were easily found in several cats from among the healthy cat population. This preliminary survey indicated that the c.1799G>A mutation was rare, suggesting that c.1799G>A might be pathogenic. In contrast, the c.55G>A mutation was not rare, and even homozygote (c.55A/A) was present without clinical symptoms among the five randomly selected clinically healthy cats. This suggested that c.55G>A was not related to feline PD with severe symptoms. However, a large-scale survey would be beneficial in detecting more cats carrying the c.1799G>A mutation and understanding the association of the c.55G>A mutation with feline cardiac function.

In human PD, arginine at amino acid position 600 is thought to be important for maintaining the function of the GAA protein because there are reports of missense mutations in which arginine at codon 600 is substituted with different amino acids, causing classical human IOPD [12,13,25,26,27,28]. These were c.1798C>T (p.R600C), c.1799G>T (p.R600L), and c.1799G>A (p.R600H) that is the same as the feline PD mutation identified in the present study. The clinical manifestations of feline PD were as severe as those in human IOPD, and feline PD caused by c.1799G>A (p.R600H) was similar to human IOPD. In human IOPD, myocardial hypertrophy due to glycogen accumulation is likely to progress to hypertrophic and dilated cardiomyopathy [29]. Therefore, both the clinical signs and histopathological findings in our cat suggest that this feline PD is an infantile form of feline PD, similar to human IOPD. There may be feline PD cases with moderate or mild cardiac disorders caused by different *GAA* mutations in the cat population.

Enzyme replacement therapy (ERT) is currently the only effective treatment of PD [30,31]. Recent studies have shown that ERT extends the survival of both children and adults. ERT has been shown to improve ventilator-free survival even in patients with IOPD. However, it is expensive because of the requirement of a large dose of GAA enzyme therapy. Furthermore, a complete cure was not evident with the present form of ERT. Therefore, improving ERT or introducing alternative therapeutic interventions, such as gene therapy, is still needed in the near future. Given the current promising achievements in the field of gene therapy, this approach deserves further investigation for the treatment of PD. The feline PD model could be instrumental in future therapeutic research as a large animal model of PD.

## 5. Conclusions

This study identified a pathogenic mutation (*GAA*:c.1799G>A, p.R600H) in the feline *GAA* gene of a DSH cat with PD. This is the first report of a cat with PD carrying the same mutation as reported in a case of human classical IOPD. The clinical and histological findings in this cat with PD were similar to those in humans with IOPD. Therefore, this feline PD is an excellent model of human PD, especially IOPD. This feline model of PD may contribute to the development of new therapeutic strategies for treating human PD.

## Figures and Tables

**Figure 1 animals-13-01336-f001:**
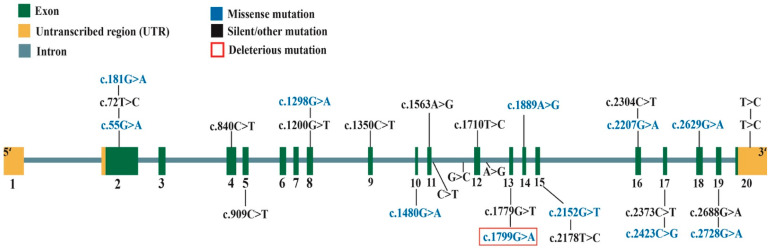
Identified variants in the *GAA* gene of the cat with Pompe disease. There were 21 homozygous coding, 2 heterozygous coding, 1 splice junction, 2 intronic, and 2 untranscribed region variants. The coding variants include 10 homozygous missense, 1 heterozygous missense mutation, and 12 silent mutations.

**Figure 2 animals-13-01336-f002:**
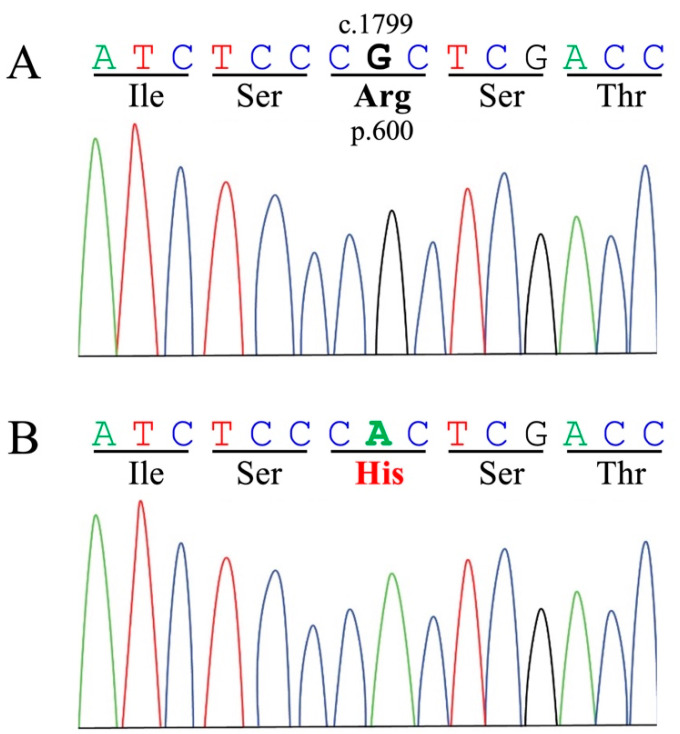
Comparison of sequenced data of the affected cat with the reference data in exon 13 of the feline *GAA* gene (c.1799G>A, p.R600H). The Sanger sequencing for a wild-type homozygous genotype (**A**) and a mutant homozygous genotype ((**B**), the affected cat) are shown.

**Figure 3 animals-13-01336-f003:**
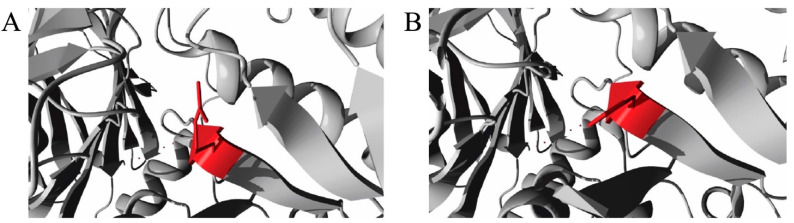
Molecular visualization of the wild-type (**A**) and the p.R600H mutant variant (**B**) of the amino acid in the feline GAA protein. The residues colored in red represent the wild-type (arginine) and variant residue (histidine) residues.

**Figure 4 animals-13-01336-f004:**
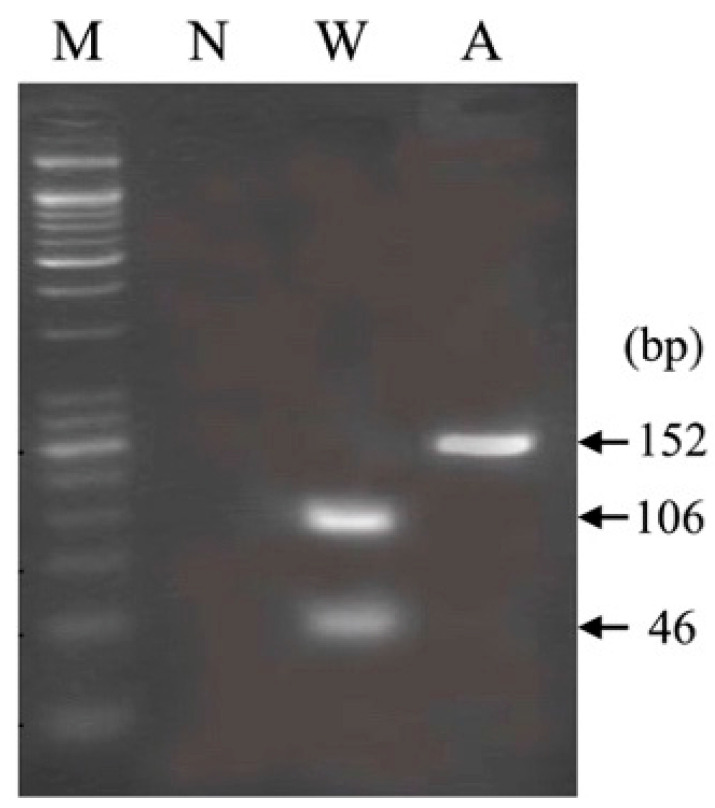
Genotyping of the wild-type (W) and affected (A) cats by polymerase chain reaction-restriction fragment length polymorphism using agarose gel electrophoresis. Lanes M and N show molecular size markers and non-template control, respectively.

**Table 1 animals-13-01336-t001:** Analysis of missense mutations in the catalytic site of the GAA protein using the SIFT and PredictSNP servers.

Exon No.	Mutations	SIFT Score	SIFT	PredictSNP	MAPP	PhD-SNP	Polyphen-1	Polyphen-2	SNAP	PANTHER
2	V19M	0.03	D	D	UK	N	D	D	D	N
2	G61S	0.92	N	N	N	N	N	N	N	UK
8	R433Q	0.91	N	N	N	N	N	N	N	N
10	E494K	0.55	N	N	N	N	N	N	N	N
13	R600H	0.00	D	D	D	D	D	D	D	D
14	E630G	0.39	N	N	N	N	N	N	N	N
15	V718L	0.32	N	N	N	D	N	N	N	N
16	R736H	0.06	N	N	N	N	N	N	N	N
17	P808R	0.24	N	N	N	N	N	N	N	N
18	V877I	0.21	N	N	N	N	N	N	N	N
19	A910T	0.60	N	N	N	N	N	N	N	N

D: deleterious; N: neutral; UK: unknown.

**Table 2 animals-13-01336-t002:** Classification of two candidate deleterious missense mutations in the catalytic site of GAA protein using iStable server.

Exon No.	Mutations	i-Mutant2.0 SEQ	DDG	MUpro	Conf. Score	iStable	Conf. Score
2	V19M	Decrease	−1.81	Null	Null	Decrease	0.570209
13	R600H	Decrease	−1.63	Decrease	−0.30256	Decrease	0.685774

## Data Availability

Not applicable.

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
