# Peer review of "Novel Mutation in the Feline GAA Gene in a Cat with Glycogen Storage Disease Type II (Pompe Disease)"

_animals, 2023, doi:10.3390/ani13081336_

Round 1

Reviewer 1 Report

This report is showing that the affected cat was found to be homozygous for the 1799G>A mutation resulting in an amino acid substitution (p.R600H) of acid α-glucosidase by 20 exons of the feline GAA gene, using genomic DNA extracted from paraffin-embedded liver tissues. It is the first report of a pathogenic mutation in the feline GAA gene. However, it remains some uncertain points in this manuscripts. 

Major Comments;

Results

Line 132: The coding variants included 10 homozygous missense, one heterozygous missense mutation, and 12 silent mutations.

→ As far as I can see, 11 missense mutation (Blue) and 11 silent mutations (Black) in Figure 1. Author should be confirm that the C.2728G>A.

Minor Comments;

Line 188-190: Sanger sequencing on five control DNA samples, and found that three homozygous (c.55G/G), two heterozygous (c.55G/A), and one homozygous mutant (c.55A/A) genotypes.

What does control means.  Five control DNA samples means are five DNA sample among 100 clinically healthy cats.?

Reviewer 2 Report

This is a beautifully prepared manuscript describing detection and in silico validation of a GAA mutation causing PD in a DSH cat. I have only one concern that the editors should address with the authors. Current usage is to refer to DNA sequence changes as "variants", reserving the word " mutation" for a variant demonstrated to cause disease.

Very minor changes:

Line 46; add the word "human" before GAA to avoid confusion.

Line 48; add a comma after 17q25.3. This would be a good place to indicate the GAA location in the feline karyotype.

Lines 249 and 250; this awkward sentence could be, "This is the first report of a cat with PD and the same mutation as reported in a case of human classical IOPD."

Reviewer 3 Report

The present paper reports the finding of a deleterious mutation in GAA genes in a cat affected by Glycogen storage disease type II. In my opinion, the study is of great interest for the scientific community, particularly from a veterinary and human medicine point of view. Overall, I found the paper to be well-written and clearly presented. I only have a few minor suggestions to improve the paper further.

Firstly, it would be helpful if you could provide some quality parameters for the PCR and sequencing methods used in the study. Since the primer pairs were not designed on the sample and the DNA was modestly fragmented, it would be informative to report some quality metrics to assess the accuracy and reliability of the results.

Secondly, it would be beneficial to have a clearer explanation as to why the c.55G>A mutation was evaluated in only 5 controls.  

Lastly, I am curious to know if it would be possible to obtain samples and/or information about the cat's parents and siblings or to collect samples from previous cases of PD. This could provide additional insights into the genetic basis of the disease and could help to further validate the findings of the study.

Finally, as a matter of curiosity, it would be interesting to know if there is any possibility to obtain samples and/or information about the cat's parents and siblings, or to collect samples from the previous case of the disease.  
